# Is Deep Remission the Right Time to De-Escalate Biologic Therapy in IBD? A Single-Center Retrospective Study

**DOI:** 10.3390/biomedicines13081928

**Published:** 2025-08-07

**Authors:** Tamara Knezevic Ivanovski, Marija Milic Perovic, Bojan Stopic, Olga Golubovic, Djordje Kralj, Milos Mitrovic, Slobodan Sreckovic, Ana Dobrosavljevic, Petar Svorcan, Srdjan Markovic

**Affiliations:** 1Department of Gastroenterology and Hepatology, University Hospital Medical Center “Zvezdara”, 11000 Belgrade, Serbia; olgaodanovic@gmail.com (O.G.); drkraljdjordje@gmail.com (D.K.); drmiloshmitrovic@gmail.com (M.M.); luckybg3@gmail.com (S.S.); svorcanp@mts.rs (P.S.); 2Department of Pathology, University Hospital Medical Center “Zvezdara”, 11000 Belgrade, Serbia; mariemilic011@gmail.com; 3Department of Nephrology, University Hospital Medical Center “Zvezdara”,110000 Belgrade, Serbia; bojanstopic@gmail.com; 4Faculty of Medicine, University of Belgrade, 11000 Belgrade, Serbia; anadobrosavljevic68@gmail.com; 5University Hospital Center Dragisa Misovic, 11000 Belgrade, Serbia

**Keywords:** mucosal healing, ulcerative colitis, Crohn’s disease, de-escalation therapy

## Abstract

**Background and Aim**: Long-term treatment with biologic therapy alongside immunomAfodulators in patients with inflammatory bowel disease (IBD) can be associated with severe side effects. The objective of this study was to determine whether discontinuing anti-TNF treatment after two years in patients who have achieved mucosal healing is associated with lower relapse rates. **Materials and Methods**: A total of 67 patients with IBD from a single tertiary IBD Center who had achieved mucosal healing were enrolled in this retrospective study. In this single-center retrospective study (January 2014–December 2022), we screened 67 IBD patients in deep remission (endoscopic mucosal healing after ≥2 years of anti-TNF therapy). After excluding three patients without histologic data, 64 patients (25 ulcerative colitis, 39 Crohn’s disease) were analyzed. Mayo endoscopic sub-score and SES-CD were used to evaluate endoscopic activity after two years of anti-TNF therapy. Histological activity was assessed using the GHAS (for CD) and Nancy index (for UC). **Results**: A total of 67 patients were screened, of whom 3 were excluded due to a lack of biopsies. Of the 64 included patients, 39.06% (25/64) had UC and 60.9% (39/64) had CD, with a mean disease duration of 11.6 ± 8.0 years. All patients were in endoscopic remission at the time of therapy de-escalation, and 60.9% (39/64) also achieved histological remission (“deep remission”). In the follow-up of 38.6 months (IQR 30–48) after biologic therapy was stopped, 57.8% (37/64) relapsed with a median time to relapse of 13.5 months (IQR 8–24) off anti-TNF—a total of 34 patients required a restarting of biologic therapy. Using Spearman’s correlation, a moderate connection was observed between histological activity at withdrawal and subsequent relapse (rho = 0.467, *p* < 0.001). The probability of relapsing within 4 years after anti-TNF cessation was significantly higher (OR 2.72) in patients with histologically active disease at the time of de-escalation. **Conclusions**: Achieving ‘deep remission’ (clinical, endoscopic, and histological healing) may be a suitable parameter for making decisions on when to de-escalate therapy; however, given that over half of patients in endoscopic remission relapse after discontinuation, any de-escalation should be approached with caution and individualized patient assessment.

## 1. Introduction

Treatment goals in inflammatory bowel disease (IBD) have evolved significantly over the past few years. The traditional step-up approach aiming for gradual control of inflammation has been replaced by early intervention with immunomodulators (IM) and/or biologics (especially anti-TNF agents), leading to tight control of inflammatory activity and timely therapy adjustments if needed (“treat-to-target”) [1,2]. Thus, the target of therapy is not only clinical remission but also mucosal healing (MH) and, in the past few years, deep remission, which is defined as no objective sign of gastrointestinal inflammation or structural abnormality [3]. Whether patients achieving deep remission can safely stop immunosuppressive therapy remains unclear. However, according to available data, deep remission is associated with fewer flares, lower surgery rates, and reduced hospitalizations [4,5]. Although the pivotal role of biologic drugs (anti-TNF agents) in achieving clinical remission, MH, and deep remission has been recognized, in clinical practice, de-escalation must be considered due to drug cost and the cumulative risks of infection and other adverse events—especially in specific situations such as pregnancy, advanced age, or significant comorbidities [6,7]. However, relapse rates after de-escalation remain high, emphasizing the need for individualized treatment decisions.

Most studies report relapse rates of 40–50% over a 2-year period following discontinuation of anti-TNF therapy [4]. For instance, the STORI trial—a multicenter prospective study assessing relapse risk after withdrawing maintenance anti-TNF in Crohn’s disease (excluding perianal disease)—found relapse rates of 43.9% at 12 months and 52.2% at 24 months, with a median time to relapse of 16.4 months [8]. In ulcerative colitis, a multicenter retrospective cohort of 193 patients on infliximab in clinical remission for ≥12 months reported an overall relapse rate of 34.7%. Among those who discontinued infliximab (*n* = 111), the 12-month relapse rate was 47.7%, compared to 17.1% in patients who continued biologic therapy (*n* = 82) [9]. Similarly, a recent observational study reported cumulative relapse incidences of 47% at 12 months and 58% at 36 months after anti-TNF withdrawal, whereas patients who maintained biologic therapy had significantly lower relapse rates (10% at 12 months; 18% at 36 months) [10]. Deep remission at the time of withdrawal appears to improve outcomes, but relapse still occurs in a substantial proportion of cases. Reported relapse rates among patients in deep remission range from ~22% to 74% at 12 months and ~50% by 36 months in various studies [4,5]. For example, Bots et al. observed an overall relapse rate of 55% after discontinuation (median time to relapse 32 months in Crohn’s and 18 months in UC, with an overall median of 28 months) [11]. Emerging evidence also suggests that persistent microscopic inflammation, even in the absence of endoscopic lesions, is associated with higher risks of relapse, hospitalization, colectomy, and neoplastic transformation [12,13]. Several factors have been identified that may predict relapse after biologic withdrawal, including male sex, younger age at diagnosis, smoking status, lack of prior surgical resection, elevated leukocyte count, lower hemoglobin, elevated C-reactive protein, and high fecal calprotectin at the time of de-escalation [14]. These considerations have been highlighted in the recent European Crohn’s and Colitis Organization (ECCO) guidelines [15].

The ongoing development of novel biologic and small-molecule therapies for IBD (e.g., anti-integrin agents, anti-IL-12/23p40 like ustekinumab, selective anti-IL-23p19 such as risankizumab and mirikizumab, and Janus kinase inhibitors) has significantly improved our ability to achieve and maintain remission [3,16]. By targeting key inflammatory pathways (for example, neutralizing TNF-α or blocking leukocyte trafficking), these therapies can induce mucosal healing and potentially alter the disease course by reducing cumulative bowel damage and long-term complications [2,17]. This paradigm shift toward early aggressive treatment raises new questions, including whether such therapies should be administered indefinitely or if they can be safely withdrawn once deep remission is achieved. Balancing the benefits of sustained therapy against risks such as infections, malignancy, and healthcare costs is crucial [7]. IBD is a lifelong disease, and a key challenge is to achieve durable remission while minimizing prolonged immunosuppression, thereby reducing risks and preserving future treatment options (including successful re-treatment with the same agent if needed). Therefore, identifying optimal timing and criteria for biologic de-escalation is of great clinical interest.

In this context, our primary goal was to evaluate whether the strategy of stopping biologic treatment in patients who have achieved mucosal healing is associated with lower relapse rates. The secondary objective was to determine if the presence of residual histological activity (as opposed to histologic remission) at the time of de-escalation predisposes patients to earlier relapse after therapy discontinuation.

## 2. Materials and Methods

Sixty-seven patients with IBD, both UC and CD, from a single tertiary IBD Center were enrolled in this retrospective study in the period from January 2014 to December 2022. The diagnosis of UC and CD was based on standard clinical, laboratory, radiological, and histological criteria in accordance with the requirements of the European Association for Crohn and Colitis (ECCO) [4,18].

Inclusion criteria were as follows:Continuous use, of at least one year, of anti-TNF therapy (infliximab and adalimumab);Mucosal healing as an endoscopic finding at de-escalation;Discontinuation of therapy as a decision of the National Committee for Biologic Therapy due to achieving clinical and endoscopic remission.

Exclusion criteria were as follows:

Endoscopic and/or clinically active disease;Decision of the National Committee to continue therapy in addition to achieving mucosal healing after two years of treatment.

Evaluation of disease activity:

Endoscopic activity in patients with CD was evaluated by SES.Mucosal healing was defined as ≤2 [19];Endoscopic activity was defined by an endoscopic score > 2 [19].Clinical activity in patients with UC was evaluated by the modified Truelove Witts score (MTWS):Clinical activity of the disease was defined as MTWS > 3 [20];Clinical remission of the disease was defined as MTWS ≤3 [20].Endoscopic activity in patients with UC was evaluated by the Mayo sub-score.Mucosal healing was defined as ≤1 [21];Endoscopic activity was defined by an endoscopic score > 1 [10].Clinical activity in patients with CD was evaluated by the CDAI score.Clinical activity of the disease was defined as CDAI > 150 [18];Clinical remission of disease was defined as CDAI ≤ 150 [18].Histological activity in patients with UC was evaluated by Nancy’s index.Histologic remission was defined by Nancy’s index ≤ 1 [22];Histologic activity was defined by Nancy’s index > 1 [22].Histological activity in patients with CD was evaluated by GHAS (Global Histologic Disease Activity Score).Histologic remission was defined by GHAS ≤ 4 [23];Histologic activity was defined by GHAS > 4 [23].

All data were collected through patients’ charts, and patients were followed for a 5-year period after discontinuation of anti-TNF agents. Histological Evaluation was performed by an experienced IBD pathologist.

## 3. Statistics

Statistical Analysis: Statistical analysis was carried out using SPSS version 20.0 (IBM, Chicago, IL, USA). Descriptive statistics for continuous variables are presented as mean ± standard deviation (SD) or median with interquartile range (IQR) as appropriate based on data distribution. Categorical variables are presented as frequencies and percentages. The normality of distributions was assessed using the Shapiro–Wilk test. Group comparisons for continuous variables were made using *t*-tests or Mann–Whitney U tests, and categorical variables were compared using the chi-square or Fisher’s exact test, as appropriate. Kaplan–Meier survival analysis was used to estimate time to relapse after therapy discontinuation, and differences between groups (e.g., stratified by disease type or histological status) were evaluated with the log-rank (Mantel–Cox) test. Spearman’s rank correlation coefficient (*rho*) was employed to assess the association between histological activity scores and relapse time (since histology scores were not normally distributed). A two-sided *p*-value < 0.05 was considered statistically significant for all analyses. No multivariable regression was performed due to the limited sample size; thus, results are based on univariate analyses and should be interpreted with caution regarding potential confounding factors.

## 4. Results (Patient Outcomes After Biologic De-Escalation)

a.Study Population and Baseline Characteristics: A total of 67 IBD patients met the initial inclusion criteria. Of these, 3 patients were excluded due to unavailable biopsy data, yielding 64 patients for analysis. Baseline demographic and disease characteristics are summarized in Table 1. The cohort included 25 UC patients (39.1%) and 39 CD patients (60.9%). The average age was 45 (SD ± 11.7) years, with 45.3% (29/64) of patients being male. The mean disease duration prior to de-escalation was 11.6 ± 8.3 years. All patients had been treated with anti-TNF therapy for a minimum of 24 months (by inclusion criteria) and had achieved endoscopic remission at the time of biologic withdrawal. The majority of patients (54.7%, 35/64) had received infliximab as their biologic, while 34.4% (22/64) had received adalimumab; a small subset (7.8%, 5/64) had been treated with both infliximab and adalimumab sequentially, and 3.1% (2/64) had been on a different advanced therapy (one on vedolizumab and one on a JAK inhibitor) prior to de-escalation. At the point of de-escalation, more than half of the patients—60.9% (39/64)—had achieved deep remission, defined as concurrent endoscopic and histological remission. After discontinuation of the biologic, 79.6% (51/64) of patients were maintained on immunomodulator therapy (thiopurine or methotrexate), whereas the remainder either received mesalamine only or no maintenance therapy. Among the 64 patients included in the cohort, 25 (39.1%) were diagnosed with ulcerative colitis (UC), the majority of whom (92%) presented with an extensive disease phenotype. The remaining 39 patients (60.9%) were diagnosed with Crohn’s disease (CD). Within the CD group, the most common disease location according to the Montreal classification was ileocolonic (L3), observed in 27 patients (69.2%), followed by colonic (L2) in 8 patients (20.5%), ileal (L1) in 4 patients (10.3%), and upper gastrointestinal involvement (L4) in 1 patient (2.5%). Regarding disease behavior, a stricturing phenotype (B2) was seen in 23 patients (58.9%), an inflammatory phenotype (B1) in 14 patients (35.8%), and a penetrating disease (B3) in 2 patients (5.1%). Additionally, perianal disease was documented in 9 patients (23.1%).

b.Clinical Outcomes After Biologic Discontinuation: The median follow-up period after stopping the biologic therapy was 39 months (IQR 30–48 months). During this time, 57.8% of patients (37/64) experienced a clinical relapse of the disease (defined as recurrence of symptoms with objective inflammation requiring treatment re-initiation). The median time to relapse was 13.5 months (IQR 8–24 months) after biologic discontinuation (Table 2). Among the 37 patients who relapsed, 13.5% (5/37) were not receiving any maintenance therapy at the time of relapse, and an additional two ulcerative colitis patients (out of the 37 relapsers) were on mesalamine alone. The remaining 30 relapsing patients (30/37, 81.1%) had been on immunomodulator maintenance (azathioprine or methotrexate) when they relapsed. No significant differences were observed between patients who relapsed and those who remained in remission in terms of baseline age (*p* = 0.620), disease duration (*p* = 0.084), duration of anti-TNF therapy (*p* = 0.110), hemoglobin level (*p* = 0.323), or C-reactive protein (*p* = 0.424) at time of de-escalation.

Kaplan–Meier survival analysis of time to relapse after biologic de-escalation is shown in Figure 1. The median relapse-free survival time for the entire cohort was 12 months after stopping biologic therapy. The survival curve demonstrates a steady decline in relapse-free probability over time, indicating an ongoing risk of relapse throughout the follow-up period. Notably, when stratified by histological remission status at the time of de-escalation, patients who had achieved histological remission (deep remission) tended to have a longer time to relapse compared to those with residual histological inflammation. Among patients with baseline histological remission, the median time to relapse was also 12 months (IQR 8–24), whereas patients without histological remission did not reach a median time to relapse within the follow-up period (because >50% of that subgroup had relapsed by the end of follow-up). By the end of follow-up, the relapse rate was markedly higher in patients who had histological activity at withdrawal (81.8% relapsed) compared to those in histological remission (46.2% relapsed). These findings suggest that the absence of histological inflammation at the time of biologic cessation is associated with a lower and delayed risk of relapse. Figure 2 illustrates the Kaplan–Meier curves stratified by histological remission status.

In our cohort, relapse-free survival did not differ significantly between patients with complicated versus non-complicated Crohn’s disease behavior. The median time to relapse was 12 months (95% confidence interval [CI] 10.0–14.0) in the complicated behavior group and 18 months (95% CI 2.9–33.1) in the non-complicated group. Kaplan–Meier curves for the two groups were similar (Figure 1), and the difference in relapse-free survival was not statistically significant (log-rank χ^2^ = 0.675, *p* = 0.411). Thus, patients with stricturing or penetrating (“complicated”) disease behavior showed a trend toward earlier relapse (median ~6 months sooner), but this did not reach significance (Figure 3).

c.Biologic Re-Initiation and Treatment Responses

Overall, 57.8% of patients (37/64) relapsed during follow-up. There was no statistically significant difference in relapse rates between ulcerative colitis and Crohn’s disease patients (57.7% vs. 58.9% relapsed, *p* = 0.77). Relapse rates also did not differ significantly by patient sex (*p* = 0.160). A prior history of surgical resection was not significantly associated with relapse (*p* = 0.643). The type of biologic therapy the patient had received (infliximab vs. adalimumab vs. other) did not considerably influence relapse risk (*p* = 0.433). Among the 37 relapsing patients, 3 experienced only mild disease activity after discontinuation (manageable with mesalamine or a short course of steroids) and thus did not require biologic re-initiation. The remaining 34 relapsing patients (91.9% of relapses) required restarting biologic therapy. Of these 34 patients, 27 (79.4%) resumed the same anti-TNF agent they had been on previously, 2 patients (5.9%) switched to a different anti-TNF agent (within-class switch, e.g., from infliximab to adalimumab or vice versa), and 5 patients (14.7%) were started on a different biologic class (such as tofacitinib, vedolizumab, ustekinumab) upon relapse. Importantly, among the patients who did not relapse (*n* = 27), only 2 (7.4%) had no maintenance therapy after de-escalation (one UC and one CD patient), while the vast majority of non-relapsers (70.4%, 19/27) remained on immunomodulator therapy (azathioprine or methotrexate). The Kaplan–Meier survival distributions regarding the association between histological activity and relapse, were compared across the different prior biologic treatment groups (IFX vs. ADA vs. sequential IFX/ADA vs. other) and showed no significant differences (log-rank *p* = 0.515, chi-square = 2.286, df = 3).

d.Histological Activity and Relapse: Histological outcomes at baseline were strongly associated with relapse patterns. Among the 37 patients who relapsed, 18 patients (48.6%) had been in histologic remission at the time of biologic withdrawal, whereas the remaining 19 relapsers (51.4%) had signs of histological activity on their last biopsy despite endoscopic healing. Notably, all of these patients with baseline histologic activity were on immunomodulator maintenance therapy after de-escalation (i.e., histologic inflammation occurred despite concurrent IM). In contrast, of the 27 patients who did not experience relapse during the follow-up period, only 3 patients (11.1%) had residual histological activity at baseline. Two of those three non-relapsing patients with baseline histologic inflammation had not been on any maintenance therapy after biologic withdrawal (which may have contributed to their ongoing microscopic inflammation). Overall, histological remission was achieved in 65.6% of the cohort (42/64 patients) at the time of biologic cessation. There was a clear difference in outcomes: patients with any histological inflammation at withdrawal were more likely to relapse than those with complete histologic remission. We found a moderate positive correlation between the degree of histological activity (by histology scores) and the likelihood of relapse (Spearman’s *rho* = 0.467, *p* < 0.001). In a univariate analysis, the odds of relapse within the ~4-year follow-up were approximately 2.72-fold higher in patients with histologically active disease at baseline compared to those in histological remission. Stated differently, deep remission (combined endoscopic and histological remission) at the time of biologic discontinuation was associated with a more favorable course, whereas microscopic disease activity portended a higher risk of disease flare.

e.Outcomes After Biologic Re-treatment (details regarding biologic therapy re-initiation and outcomes are provided in Table 3):

In the subgroup of patients who relapsed and had at least 4 years of follow-up after biologic discontinuation (*n* = 28, a subset with extended follow-up), 64.3% (18/28) ended up resuming biologic therapy. Among the 34 total patients who restarted biologics (from the entire relapsing group), most (27/34, 79.4%) were re-induced with the same anti-TNF agent to which they had previously responded, while 2/34 (5.9%) switched to a different anti-TNF within the class, and 5/34 (14.7%) switched to a biologic with a different mechanism of action (outside the anti-TNF class). Outcomes after re-initiation were generally favorable: of the patients who restarted biologic therapy and were followed up for response, 15/18 (83.3%) achieved remission after re-induction of treatment. However, 2 of 18 patients (11.1%) failed to respond to reintroduced biologic treatment and ultimately required surgical intervention (resection). One patient who was restarted on the same anti-TNF agent did not respond initially; after switching to an alternative anti-TNF agent in the same class, that patient did achieve remission. Two patients (out of the eighteen with documented outcomes) experienced serious adverse events upon restarting anti-TNF therapy: one had an anaphylactic infusion reaction and another developed severe psoriasis, necessitating discontinuation of the anti-TNF. These cases underscore that while re-treatment is often effective, it is not without risk and occasional failure.

It is notable that among the patients who did not relapse after anti-TNF withdrawal (*n* = 27), only 2 (7.4%) had received no medical maintenance therapy, and both of those patients remained relapse-free (one with CD and one with UC). The rest of the non-relapsers were largely maintained on an immunomodulator (70.4%) or, in a few cases, mesalamine. This observation suggests that ongoing background therapy (even non-biologic) may have contributed to sustaining remission in some patients, although our study was not designed to compare maintenance strategies formally. A log-rank (Mantel–Cox) analysis found no significant difference in relapse-free survival when comparing subgroups based on the class of biologic initially used (IFX vs. ADA vs. other; *p* = 0.515), consistent with relapse risk being more strongly associated with disease biology (e.g., histologic activity) than the specific anti-TNF agent.

## 5. Discussion

The ongoing development of novel agents for the treatment of inflammatory bowel disease (IBD) has significantly improved the ability to achieve and maintain remission, marking a paradigm shift toward more effective disease management. However, a top-down approach to therapy, which prioritizes the early and aggressive use of these advanced agents to achieve rapid disease control, raises new questions. These include whether such treatments can be administered indefinitely or should be discontinued at a certain point due to considerations such as patient preferences, potential adverse side effects, secondary loss of response, health policy constraints, or financial limitations. This highlights the need for a balanced strategy that integrates clinical evidence, patient-centered care, and long-term health system sustainability [6]

Most studies report relapse rates ranging from 40% to 50% over a 2-year period following the discontinuation of anti-TNF therapy [4]. For instance, the STORI trial, a multicenter, prospective study specifically designed to assess the risk of relapse following the withdrawal of anti-TNF maintenance therapy in patients with luminal Crohn’s disease (CD) excluding perianal disease, observed relapse rates of 43.9% ± 5.2% at 12 months and 52.2% ± 5.2% at 24 months, with a median time to relapse of 16.4 months [9]. For ulcerative colitis, a multicenter retrospective cohort study involving 193 patients treated with infliximab reported a relapse rate of 34.7% among those in clinical remission for a minimum of 12 months. Among patients who discontinued infliximab (*n* = 111), the relapse rate was 47.7% compared to 17.1% in those who continued biologic treatment (*n* = 82) [23].

Similar findings have been observed in studies involving patients in deep remission (clinical and endoscopic remission), with relapse rates ranging from 22% to 74% at 12 months, 47% to 49% at 24 months, and approximately 50% at 36 months [24,25]. A recent study by Bots et al. reported a relapse rate of 55% during the observation period. Among these, 49% of CD patients relapsed with a median time of 32 months, while 75% of UC patients relapsed with a median time of 18 months. The overall median time to relapse across all patients was 28 months [26].

In our cohort, we observed a relapse rate of 57.8% following anti-TNF discontinuation, aligning with rates reported in prior studies, including the STORI trial and other cohorts [9]. Importantly, we identified histological activity as a moderate predictor of relapse risk, underscoring the potential value of incorporating histologic remission as a criterion when considering de-escalation. This finding supports emerging evidence that microscopic inflammation, even in the absence of clinical or endoscopic activity, can predict future disease flares and may guide treatment decisions.

In a multicenter retrospective cohort study with 193 patients, which included cases of UC treated with infliximab and those in clinical remission for a minimum of 12 months, the overall relapse rate was 34.7%. Among patients who discontinued infliximab [23], Casanova et al. assessed the relapse risk of 1055 patients who discontinued anti-TNF drugs after achieving clinical remission with a median follow-up of 19 months. The cumulative incidence of relapses was 44% per patient-year (15% at 6 months, 24% at 1 year, 38% at 2 years, 46% at 3 years, 56% at 5 years), without significant differences between CD and UC patients (*p* = 0.1) [14].

Persistent microscopic inflammation has been consistently associated with increased relapse rates, hospitalization, colectomy risk, and even neoplastic transformation. Several studies have shown that persistent microscopic inflammation beyond clinical and microscopic remission is associated with increased relapse rates, hospitalization, colectomy, and risk of neoplasm [27,28]. Other analyses have highlighted factors that predict the risk of relapse after anti-TNF cessation, including male sex, age at diagnosis, smoking status, the absence of surgical resection, a leucocyte count greater than 6.0 × 10^9^/L, a hemoglobin level of ≤145 g/L, a C-reactive protein level of ≥5.0 mg/L, and elevated fecal calprotectin [12,29].

Our finding of comparable relapse-free survival between complicated and non-complicated Crohn’s disease behavior is in line with specific reports in the literature, though overall evidence is mixed. On one hand, problematic behavior, particularly penetrating disease, has been traditionally associated with a more aggressive course. For example, one cohort of quiescent Crohn’s patients showed that penetrating disease conferred an elevated relapse risk (hazard ratio ~3.7), and such patients are often noted to have higher rates of surgery and postoperative recurrence [30]. On the other hand, some studies have found no clear impact of baseline behavior on relapse; in an extensive analysis of anti-TNF withdrawal, penetrating vs. non-penetrating disease did not significantly affect relapse rates (HR ~0.95, 95% CI 0.76–1.18) [31]. Our results resemble the latter, suggesting that disease behavior alone may not determine short-term relapse risk. This implies that other factors—such as treatment strategies, disease location, and early intervention—could mitigate the expected disadvantages of complicated behavior, underscoring the importance of comprehensive management beyond phenotypic classification.

The strength of our study lies in its significant findings, as all patients achieved both clinical and endoscopic remission, which highlights the effectiveness of the therapeutic approach. Additionally, most patients were transitioned to immunomodulator therapy after discontinuation of anti-TNF treatment, providing valuable insights into potential maintenance strategies following biologic therapy.

However, this study also has notable limitations. One primary limitation is the small sample size, which may restrict the generalizability of our findings to the broader population. Furthermore, the retrospective nature of data collection introduces potential biases, as not all histological and biochemical data were consistently available. Key biochemical parameters, such as hemoglobin levels, leukocyte counts, and fecal calprotectin, which are known predictors of relapse, could not be systematically incorporated into the analysis. This gap limits the ability to fully assess the factors contributing to disease recurrence following treatment cessation. Another significant limitation is related to the timing and criteria for discontinuing biologic therapy. Patients were transitioned off biological treatment immediately after achieving mucosal healing, without individualized consideration of disease severity, duration, smoking status, prior surgical interventions, or the presence of perianal disease. These factors are critical in determining long-term outcomes and relapse risks but were not accounted for due to constraints imposed by the national health policy at the time. Specifically, the policy limited the duration of biologic therapy as a cost-containment measure within a restricted national healthcare budget.

These limitations underscore the need for prospective studies with larger cohorts to address these gaps. Future research should include a broader set of clinical, biochemical, and histologic parameters to identify predictors of sustained remission better. Additionally, evaluating the impact of tailored treatment discontinuation strategies, as opposed to fixed timelines, could provide more robust insights into optimizing patient outcomes while balancing healthcare resource utilization.

Future research should focus on prospective studies with standardized monitoring, including histologic, endoscopic, and biomarker assessments, to develop predictive models for relapse risk and to establish clear criteria for safe biologic de-escalation in IBD management.

## 6. Conclusions

In conclusion, our data provide evidence to support the notion that deep remission (i.e., including mucosal and histological remission) may be a valuable parameter to consider when re-evaluating the possibility of IBD biologic therapy de-escalation. Patients without endoscopic or histologic activity were at lower risk of relapsing after ATI, so deep (as opposed to clinical) remission should be an attractive condition for withdrawal. But it is not a promise of ongoing remission: Of the patients, 57.8 percent relapsed, many of them from deep remission. Hence, de-escalation decisions must be made cautiously and individualized, considering factors such as age, disease phenotype, biomarker levels (e.g., fecal calprotectin), and risk tolerance. Close monitoring and a readiness to reinitiate therapy are essential. While deep remission is a promising guideline, it should be part of a broader decision-making framework. Larger prospective studies with long-term follow-up are required to address its prognostic value and the long-term consequences of discontinuing treatment.

## Figures and Tables

**Figure 1 biomedicines-13-01928-f001:**
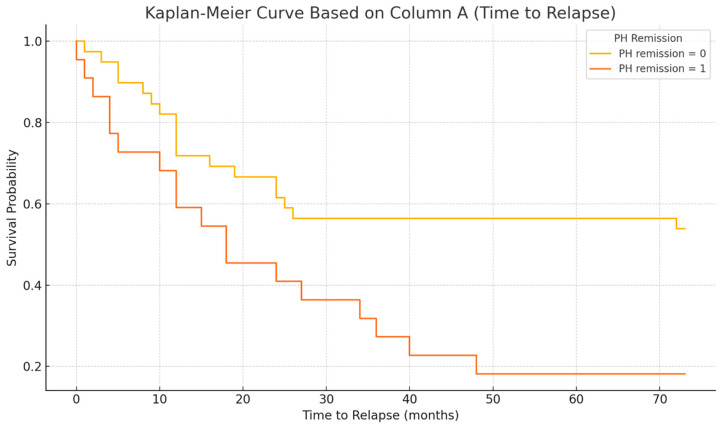
Kaplan–Meier curve showing time to relapse in all patients after biologic therapy de-escalation (anti-TNF withdrawal). The median time to relapse was 12 months post-discontinuation, and the cumulative probability of remaining relapse-free declined over the 4-year follow-up period.

**Figure 2 biomedicines-13-01928-f002:**
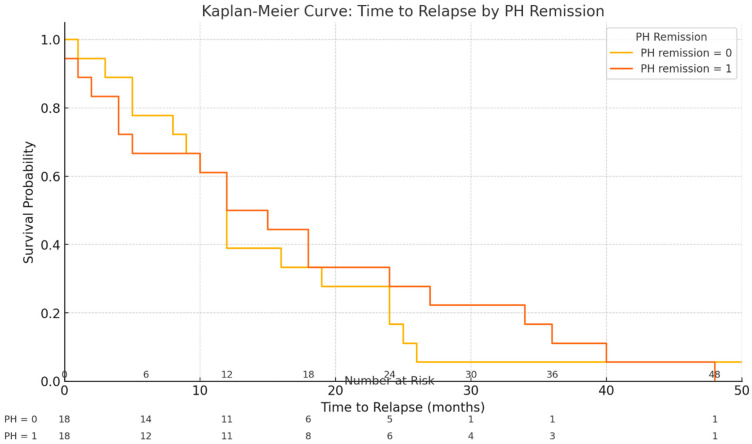
Kaplan–Meier curves for time to relapse stratified by histological remission status at the time of biologic withdrawal. Patients in histological remission (deep remission) at de-escalation demonstrated a lower risk and longer latency to relapse compared to those with histological activity. No median time to relapse was reached in the histologically active group within 48 months, reflecting their higher relapse rate.

**Figure 3 biomedicines-13-01928-f003:**
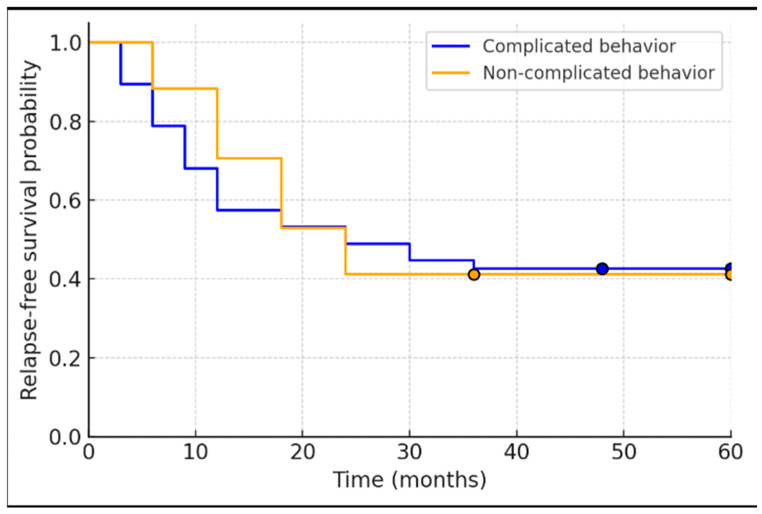
Kaplan–Meier relapse-free survival curves for Crohn’s disease patients stratified by disease behavior (complicated vs. non-complicated). The curves are closely overlapping, reflecting no significant difference between groups (log-rank *p* = 0.41). Median relapse-free survival was approximately 12 months in the complicated behavior group and 18 months in the non-complicated group, with substantial overlap in their confidence intervals.

**Table 1 biomedicines-13-01928-t001:** Patient demographic and baseline disease characteristics.

Characteristics	Value (N = 64)
Male, *n* (%)	29 (45.3%)
Age (mean ± SD), years	45.1 ± 11.8
Disease duration (mean ± SD), years	11.6 ± 8.3
Diagnosis: Ulcerative colitis (UC) Extensive distal	25 (39.1%) 23 (92%) 2 (8%)
Diagnosis: Crohn’s Disease (CD) L1 4 L2 8 L3 27 L4 1 B1 14 B2 23 B3 2 P 9	39 (60.9%) 4 (10.25%) 8 (20. 51%) 27 (69.2%) 1 (2.5%) 14 (35.8%) 23 (58.9%) 2 (5.1%) 9 (23.07%)
Biologic agent at withdrawal: Infliximab (anti-TNF)	35 (54.7%)
Biologic agent: Adalimumab (anti-TNF)	22 (34.4%)
Biologic agent: Sequential IFX → ADA	5 (7.8%)
Biologic agent: Other (vedolizumab or JAK inhibitor)	2 (3.1%)
Histological remission at withdrawal (deep remission)	39 (60.9%)
On immunomodulator maintenance post-biologic, *n* (%)	51 (79.7%)

**Table 2 biomedicines-13-01928-t002:** Follow-up outcomes after biologic cessation.

Follow-Up Outcomes	Value
Follow-up duration, median (IQR)	39.0 months (30–48)
Median time to relapse (among relapsers)	13.5 months (8–24)
Maintained immunomodulator therapy	51/64 patients (79.6%)
Patients with clinical relapse	37/64 patients (57.8%)
No maintenance therapy at relapse	5/37 relapsers (13.5%)
Mesalamine only at relapses (UC)	2/37 relapsers (5.4%)
Immunomodulator at relapse (AZA/MTX)	30/37 relapsers (81.1%)

**Table 3 biomedicines-13-01928-t003:** Biologic therapy re-initiation and outcomes in relapsing patients.

Restart of Biologic Therapy	Patients (*n*)
Patients who restarted biologics after relapse	34
– Resumed the same anti-TNF agent	27/34 (79.4%)
– Switched to a different anti-TNF (class swap)	2/34 (5.9%)
– Switched to a different class (e.g., vedolizumab, ustekinumab)	5/34 (14.7%)
Outcome after biologic re-initiation (subset analyzed, *n* = 18)	
– Achieved remission after restart	15/18 (83.3%)
– Primary non-response (required surgery)	2/18 (11.1%)
– Initial non-response but remission after switch within class	1/18 (5.6%)
– Serious adverse event (e.g., anaphylaxis, psoriasis)	2/18 (11.1%)

## Data Availability

All data supporting the findings of this study are contained within the article. Further inquiries can be directed to the corresponding authors.

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
