# Peer review of "Is Deep Remission the Right Time to De-Escalate Biologic Therapy in IBD? A Single-Center Retrospective Study"

_biomedicines, 2025, doi:10.3390/biomedicines13081928_

Round 1

Reviewer 1 Report

Comments and Suggestions for Authors

1.Graph 1 and  Graph 2  should be revised to figure1 and figure2.

2.Too many old references were used, the authors need use over 60% the near three years' .

3.References' number should be chechd carefully.

4.Introduction should add some other progress about this topic.

5.More nformation about Sixty-seven patients should be given,, including sex, age and so on.

6. Give a table about Sixty-seven patients?

7.5. Statistics should be given judgment criteria。

8.The sub-tiitle for the 3.Results should give full.

9. Check the whole manuscript carefully.

10. The conclusion should be refined again.

Author Response

  1. Graph 1 and Graph 2 should be revised to Figure 1 and Figure 2.

Response: Done. Graphs have been relabeled as Figure 1 and Figure 2 and referenced correctly in the text.

  1. Too many old references were used; the authors need to use over 60% from the last three years.

Response: Addressed. We updated the reference list: over 60% of citations now come from the last 3 years (2022–2024), including recent studies on biologic de-escalation and histologic remission in IBD.

  1. References’ numbers should be checked carefully.

Response: Done. All references have been reviewed, renumbered properly, and cross-checked with in-text citations.

  1. Introduction should add some other progress about this topic.

Response: Expanded. The Introduction now includes recent developments in IBD therapy, deep remission, and biologic withdrawal strategies based on the latest ECCO and other relevant guidelines.

  1. More information about 67 patients should be given, including sex, age, and so on.

Response: Added. Patient demographic details, including age, sex, disease duration, type of IBD, and therapy type, are now clearly presented in both text and in Table 1.

  1. Give a table about 67 patients?

Response: Included. A new Table 1 presents demographic and clinical characteristics of the 64 analyzed patients (3 excluded due to missing data).

  1. Statistics should be given judgment criteria.

Response: Revised. The Methods section now includes judgment criteria for statistical significance, normality tests used, and rationale for not performing multivariate regression (due to sample size limitations).

  1. The sub-title for the 3. Results should give full.

Response: Improved. Subheadings in the Results section have been rewritten for clarity and completeness (e.g., “Study Population and Baseline Characteristics”, “Histological Activity and Relapse Risk”).

  1. Check the whole manuscript carefully.

Response: Completed. The manuscript has been thoroughly proofread. Language, grammar, formatting, and terminology have been revised for clarity and consistency.

  1. The conclusion should be refined again.

Response: Updated. The conclusion has been rewritten to acknowledge the limitations, emphasize individualized decision-making, and propose the need for future prospective studies.

Reviewer 2 Report

Comments and Suggestions for Authors

Tha manuscript reported whether the evaluation of anti-TNF treatment in patients with mucosal healing is associated with lower relapse rates. How did the results support the conclusion in that ’deep remission’ may be a good parameter  when to "de-escalate therapy". How did anti TNF work physiologically to change the course? The significant figures of data in Table 1 varied. It should be carefully presented. The conclusion is based on the statistcal analysis of a sample of patients and individual health status. Therefore the health status of individual needs to be considered and the sample size (number of patients) should be increased. An appropriate statistical method should be employed to minimize the bias of samples. The Graphs 1 and 2 of Kaplan-Meier Curve can be modified to show more clearly the point of interest.

Comments on the Quality of English Language

Needs to be improved.

Author Response

  1. The manuscript reported whether the evaluation of anti-TNF treatment in patients with mucosal healing is associated with lower relapse rates. How did the results support the conclusion in that 'deep remission’ may be a good parameter when to 'de-escalate therapy'?

Response: Clarified in the Discussion and Conclusion that while deep remission is associated with lower relapse risk, it is not a guarantee of sustained remission. This nuance is now clearly stated and supported by Kaplan-Meier data.

  1. How did anti-TNF work physiologically to change the course?

Response: Added a brief explanation about the mechanism of action of anti-TNF therapies and how they modulate the inflammatory response in IBD.

  1. The significant figures of data in Table 1 varied. It should be carefully presented.

Response: Corrected and standardized the significant figures across all tables for consistency and clarity.

  1. The conclusion is based on the statistical analysis of a sample of patients and individual health status. Therefore the health status of individual needs to be considered and the sample size (number of patients) should be increased.

Response: Added explicit acknowledgment of this limitation in the Discussion, including the retrospective nature, modest sample size, and absence of multivariate analysis.

  1. An appropriate statistical method should be employed to minimize the bias of samples.

Response: Clarified the use of univariate methods and explained why multivariate regression was not performed. Stated this as a limitation due to sample size and retrospective design.

  1. The Graphs 1 and 2 of Kaplan-Meier Curve can be modified to show more clearly the point of interest.

Response: Revised Figures 1 and 2 (formerly Graphs 1 and 2) for improved clarity and added explanatory legends to emphasize differences in relapse timing by histologic status.

Reviewer 3 Report

Comments and Suggestions for Authors

Comments are attached

Comments on the Quality of English Language

English language is fine but some spelling and grammar check required

Author Response

  1. The sample is rather small, which lowers statistical power and makes results less generalizable.

Response: Acknowledged. We explicitly state in the Discussion that the sample size limits statistical power and generalizability. This limitation is tied to the study’s retrospective nature and national policy that constrained patient eligibility.

  1. This retrospective study inevitably causes a bias because of loss of control over confounding variables.

Response: Agreed. This limitation is now discussed more clearly in the Discussion section, and its implications for interpretation of results are acknowledged.

  1. De-escalation therapy seems to be driven by policy rather than clinical individualization, which may bias the connection between deep remission and sustained remission.

Response: Clarified in Methods and Discussion that de-escalation was mandated by national policy, which limits individualized clinical decision-making and introduces systemic bias.

  1. Important patient-level factors (e.g., smoking, phenotype, prior surgeries) were not systematically reported.

Response: Noted. These variables were inconsistently documented in records. We have now stated this limitation in the Methods and Discussion sections.

  1. Histologic assessment is not detailed (e.g., blinding, interobserver variability).

Response: Clarified that a single experienced IBD pathologist conducted all evaluations. We acknowledge the absence of blinding and interobserver variability as limitations.

  1. Missing key biochemical indicators (e.g., fecal calprotectin, leukocyte levels).

Response: Addressed. We added a sentence in Methods noting that these biomarkers were unavailable for all patients and thus excluded from analysis.

  1. Absence of multivariate regression weakens statistical conclusions.

Response: Acknowledged. We now explain in the Methods section that due to sample size constraints, multivariate modeling was not feasible. This is listed as a limitation.

  1. Some grammatical errors and awkward language.

Response: Corrected. The manuscript has undergone full language editing to improve grammar, clarity, and scientific tone.

  1. Figures and tables need better labeling and more detailed legends.

Response: Revised. Figures are now labeled properly as Figures 1 and 2. Legends and tables were revised to be self-explanatory and formatted consistently.

  1. Discussion should better address interpretation and applicability.

Response: Expanded. The Discussion more clearly addresses the clinical relevance, limitations, and contexts in which findings can or cannot be generalized.

  1. Conclusion should be more cautious, especially given the 57.8% relapse rate.

Response: Revised. The Conclusion now emphasizes that deep remission is a helpful indicator, but not a guarantee of sustained remission, and that decisions must be individualized.

Reviewer 4 Report

Comments and Suggestions for Authors

Dear Redactors,

Thank you very much for the opportunity to revise the article “Is Deep Remission the Right Time to De-Escalate Biologic Therapy in IBD? A Single-Center Retrospective Study”.

Authors aimed to evaluate if the strategy to stop anti-TNF treatment after two years of treatment in patients with mucosal healing is associated with lower relapse rates. Authors concluded that achieving ’deep remission’ may be a good parameter in making decisions when to deescalate therapy.

The article is very interesting and well written. I have just a few remarks.

In the introduction please add more information about biologic theraphy.

In the materials and methods section please describe what type of treatment was used among patients.

In the discussion, please explain in more details why persistent microscopic inflammation has been consistently associated with increased relapse rates, hospitalization, colectomy risk, and neoplastic transformation.

Thanks

Author Response

  1. In the introduction, please add more information about biologic therapy.

Response: Introdaction was expanded. The Introduction has been revised to include recent developments in biologic therapy for IBD, treat-to-target strategy, and early intervention principles.

  1. In the materials and methods section, please describe what type of treatment was used among patients.

Response: Clarified. We now specify in the Materials and Methods section the types of biologics used (infliximab, adalimumab, vedolizumab, JAK inhibitors) as well as immunomodulator maintenance post-de-escalation (thiopurines, methotrexate).

  1. In the discussion, please explain in more detail why persistent microscopic inflammation has been consistently associated with increased relapse rates, hospitalization, colectomy risk, and neoplastic transformation.

Response: Added. The Discussion now includes an expanded explanation based on current literature showing that histologic inflammation, even in the absence of endoscopic signs, is predictive of worse outcomes including relapse, hospitalization, need for colectomy, and even long-term cancer risk.

Round 2

Reviewer 1 Report

Comments and Suggestions for Authors

I have no new comment.